# DUAL DENOISING LOGICAL REASONING FOR INDUCTIVE KNOWLEDGE GRAPH COMPLETION

## ABSTRACT

In recent years, message-passing Graph Neural Networks (GNNs) have been widely used to tackle the problem of inductive knowledge graph completion. Though great progress has been made in GNN-based knowledge graph reasoning, it still suffers from the noise existing in irrelevant entities. These noises accumulated exponentially as the reasoning process continues, significantly impacting the overall performance of the model. To tackle this problem, several node-based sampling methods have been proposed for denoising. However, they do have inherent limitations. Firstly, they rely on node scores to evaluate node importance, which cannot effectively assess the quality of paths in GNN-based reasoning. Secondly, they often overlook noise interference caused by irrelevant edges. To address these problems, we propose a dual denoising logical reasoning (DDLR) framework, which integrates path-based and edge-based sampling to achieve comprehensive denoising. Specifically, DDLR employs a path-scoring mechanism to evaluate the importance of paths, aiming to remove irrelevant paths. Moreover, DDLR leverages rules within the knowledge graph to remove irrelevant edges. Through the dual denoising process, we can achieve more effective logical reasoning. To demonstrate the effectiveness of the DDLR framework, conduct experiments on three benchmark datasets, and our approach achieves state-of-the-art performance.

## 1 INTRODUCTION

Knowledge Graphs (KGs) are widely used to formulate the structure information among real-world entities. They employ nodes to represent entities and edges to capture their relations. Despite their utility, KGs are often afflicted by inherent incompleteness and inaccuracies. To address these limitations, the vital task of Knowledge Graph Completion (KGC) has emerged. KGC is dedicated to inferring missing or latent facts within KGs and has been extensively explored in prior research (Bordes et al., 2013; Galárraga et al., 2015; Ho et al., 2018). This task holds particular importance in various applications, such as question answering (Zhang et al., 2018), natural language processing (Zhang et al., 2019), and recommendation systems (Wang et al., 2018), where the precision and comprehensiveness of knowledge representation are paramount for achieving high-quality results.

Various methods have been developed for KGC to predict missing facts, including TransE (Bordes et al., 2013), DistMult (Yang et al., 2014), and ConvE (Dettmers et al., 2018). These methods map entities and relations to low-dimensional spaces and predict missing relations through tensor operations. However, traditional KGC models are inherently transductive, requiring the presence of all entities in the training data. Since knowledge graphs evolve over time, inductive knowledge graph completion becomes a hot topic, where new entities emerge. In such cases, traditional embedding-based models struggle to adequately represent emerging entities and perform poorly in inductive KGC.

Inductive KGC is a challenging task because new entities emerge in the testing phase, which is not seen during the training process. Therefore, how to effectively represent the new entities becomes a problem. Some works utilize GNNs to acquire meaningful representations of entities within knowledge graphs. (Teru et al., 2020) proposed GraIL, aiming to implicitly derive logical rules by analyzing the subgraph surrounding a candidate triplet in an entity-independent manner. Subsequent research efforts have enhanced this process by considering topological relations (Chen et al., 2021)

or optimizing message passing through node-edge interactions (Mai et al., 2021). NBFNet (Zhu et al., 2021) and RED-GNN (Zhang & Yao, 2022) introduce a path-based approach for direct learning of relation representations, eliminating the need to learn entity embeddings. Despite the good performance of these methods, they follow the same GNN message-passing process, i.e., propagating messages freely through all edges in graphs, which would bring in noise. In order to minimize the influence of noise, (Zhang et al., 2023a) propose to select the top K nodes in each iteration using learnable attention mechanisms. Nonetheless, there are two limitations to the existing studies. Firstly, take the knowledge graph in Fig 1 as an example, we can easily deduce that the answer to the query $(a, Mother, ?)$ is $d$ because there exists an evidence path $Father(a, b) \land Wife(b, d)$. However, according to the top-K strategy, the result obtained is $e$, because node $e$ has a higher score than node $d$. Moreover, this strategy might also deduce the correct answer through pseudo-evidence reasoning, as exemplified by the path $Friend(a, f) \land Mother(f, g) \land Friend(g, d)$. Secondly, from Fig 1, we can also intuitively observe that the relevance of various relations within the red path to the target relation is higher than the relevance of various relations within the green path to the target relation. Therefore, the correlation between relations within the path and the target relation helps the model filter out irrelevant reasoning paths (i.e., the green path). It is evident that the current methods have not taken this into account.

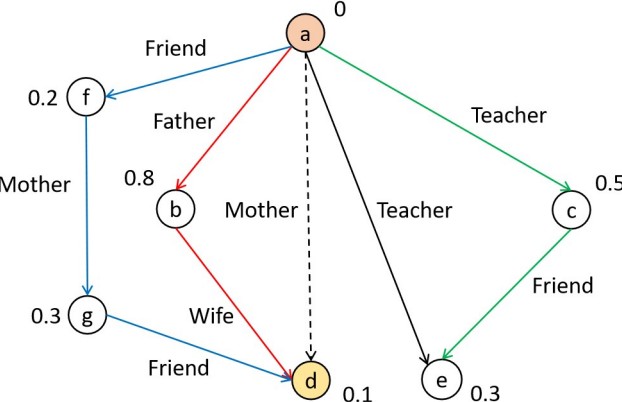

Figure 1: An example of KG reasoning, where the query is $(a, Mother, ?)$ and the answer is $d$. Green/Blue lines represent irrelevant paths (i.e., non-evidence path) to this query, while the red line is relevant (i.e., evidence path). The dashed line indicates a missing fact, and numerals represent node scores.

To address these challenges, this paper proposes a **D**ual **D**enoising **L**ogical **R**easoning (DDLR) framework, which achieves comprehensive denoising through path and edge sampling. Specifically, for path sampling, we designed a novel path-scoring mechanism, which takes into account not only the current path score but also the scores of the remaining paths, to evaluate the importance of the path for sampling paths most relevant to the objective. For edge sampling, we leverage single rules(i.e., ordered relation pairs), which reveal correlations between relations, to explore the most relevant edges while filtering out irrelevant paths. In addition, to avoid the information loss caused by the top-k strategy, we adopt a sampling method that combines the Bernoulli distribution and rule confidence to sample edges. In our experiments, we observed that this sampling strategy can preserve promising targets during the reasoning process, thus enhancing the model's performance.

To summarize, the main contributions of this paper are as follows:

- We propose a dual denoising logical reasoning model that comprehensively addresses noise interference in inductive KGC by both path sampling and edge sampling.
- We propose a path-scoring mechanism to assess the effectiveness of paths, enabling the precise selection of paths relevant to the target while simultaneously discarding irrelevant ones. This innovation enhances the reasoning capability of DDLR within knowledge graphs.
- We extract single rules at the triplet level and apply them to the edge sampling process. In contrast to conventional methods, it places a stronger emphasis on finer-grained triplets, thereby improving rule credibility and making sampling more logical.

- We conduct extensive experiments on three benchmark datasets to assess the effectiveness of DDLR. The results illustrate that our proposed algorithm surpasses state-of-the-art baselines in terms of prediction accuracy.

## 2 RELATED WORK

### 2.1 GNNS ON KNOWLEDGE GRAPHS

Significant progress (Yang et al., 2015; Schlichtkrull et al., 2018; Toutanova et al., 2015b; Dettmers et al., 2018; Socher et al., 2013; Wang et al., 2019; García-Durán et al., 2017) has been made in addressing the challenge of multiple relationship edges in knowledge graph completion through the extension of GNNs. These efforts have led to innovative message-passing operations that are carefully designed to accurately capture complex relationship information during feature transformation and aggregation processes. Prominent knowledge graph completion models include RGCN (Schlichtkrull et al., 2018), which enhances reasoning capabilities by utilizing relation-specific transformation matrices; CompGCN (Vashishth et al., 2020), which seamlessly integrates neighborhood information through entity-relation composition operations; KBGAT (Yang et al., 2015), a groundbreaking model that uses attention mechanisms to differentiate entity roles in multiple relationships; and recently introduced models in this field such as NBFNet (Zhu et al., 2021) and RED-GNN (Zhang & Yao, 2022), which introduce progressive propagation strategies from query entities to their multi-hop neighbors. These models have shown potential in both inductive and transductive reasoning scenarios and have demonstrated efficiency in evaluating multiple candidate entities. Additionally, RuleNet (Zhang et al., 2023b) has achieved remarkable results by effectively capturing potential relationship connections. However, the number of irrelevant entities involved exponentially increases with deeper propagation paths, making it more difficult to identify target answer entities.

### 2.2 SAMPLING METHODS FOR GNNS

Traditional message propagation methods face scalability issues when dealing with large-scale Graph Neural Networks (GNNs) (Hu et al., 2020). To address this challenge, several sampling methods have been proposed. For instance, GraphSAGE (Hamilton et al., 2017) and PASS (Chen et al., 2018a) sample a fixed-size neighborhood set from the complete neighbors of each node at each propagation step, while FastGCN (Yang et al., 2020), Adaptive-GCN (Chen et al., 2018b), and LADIES (Shi et al., 2019) sample an equal number of nodes at each propagation step. On the other hand, Cluster-GCN (Chiang et al., 2019), GraphSAINT (Hamilton et al., 2020), and ShadowGNN (Ma et al., 2021) extract subgraphs containing highly relevant nodes around anchor nodes. To address the challenge of convergence speed, a new subgraph sampling method with guaranteed convergence has recently been proposed (Shi et al., 2023). In contrast, the LAyer-neighBOR sampling (LABOR) (Balın & Ümit V. Çatalyürek, 2022) algorithm aims to reduce the number of sampled vertices by directly replacing neighborhood sampling with the same diffusion hyperparameters while maintaining quality (Balın & Ümit V. Çatalyürek, 2022). Adaprop (Zhang et al., 2023a) designs an incremental sampling mechanism to preserve proximate and promising targets. However, they overlook the impact of edges on sampling. In contrast, our proposed sampling strategy comprehensively considers the influence of both paths and edges on sampling, thereby improving the model's performance.

## 3 PRELIMINARY

**Knowledge Graph Completion**. A knowledge graph, denoted as $\mathcal{G} = (\mathcal{V}, \mathcal{E}, \mathcal{R})$, comprises finite sets of entities (nodes) represented as $\mathcal{V}$, facts (edges) as $\mathcal{E}$, and relation types as $\mathcal{R}$. Each fact is a triplet $(x, r, y) \in \mathcal{V} \times \mathcal{R} \times \mathcal{V}$, signifying the presence of relation $r$ from entity $x$ to entity $y$. Knowledge graph completion is a task focused on predicting answers for queries such as $(u, q, ?)$ or equivalently, $(u, q^{-1}, ?)$, where $q^{-1}$ represents the inverse relation of $q$. Given a query $(u, q, ?)$, the objective is to determine the set of answers $\mathcal{V}(u, q, ?)$ such that for all $v \in \mathcal{V}(u, q, ?)$, the triplet $(u, q, v)$ holds true.

**Path-based Approaches**. Path-based approaches achieve knowledge graph completion by analyzing the paths between a pair of entities within a knowledge graph. For example, a path

$Father(a, b) \wedge Wife(b, d)$ may be used to predict $Mother(a, d)$ in Fig 1. From a representation learning perspective, path-based approaches aim to learn a representation $\boldsymbol{h}_q(u, v)$ to predict the triplet $(u, q, v)$ based on all paths $\mathcal{P}_{u \to v}$ from entity $u$ to entity $v$.

$$\boldsymbol{h}_q(u, v) = \bigoplus_{P \in \mathcal{P}_{u \to v}} \boldsymbol{h}_q(P) = \bigoplus_{P \in \mathcal{P}_{u \to v}} \bigotimes_{(x,r,y) \in P} \boldsymbol{w}_q(x, r, y) \tag{1}$$

where $\bigoplus$ is a permutation-invariant aggregation function, $\bigotimes$ is an aggregation function. $\boldsymbol{w}_q(x, r, y)$ is the representation of triplet $(x, r, y)$ conditioned on the query relation $q$.

## 4 METHOD

In this section, we provide a detailed introduction to the DDLR framework, which employs dual denoising for more efficient logical reasoning. The framework, as shown in Fig 2, consists of several iterative sampling units. Each iterative sampling unit comprises two sub-modules: path-based sampling and edge-based sampling. DDLR employs these iterative sampling units to remain the most relevant paths or edges for the task.

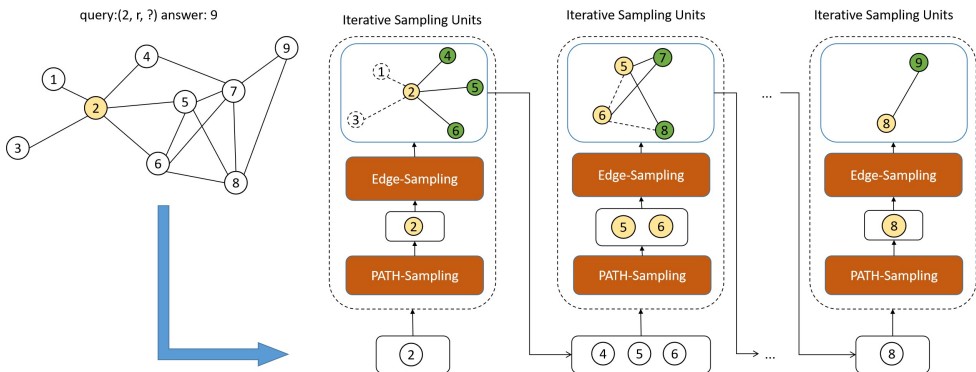

Figure 2: An overview of DDLR. For simplicity, we use circles to represent entities and ignore the types of relation. Dashed lines and dashed circles represent irrelated relations and entities, respectively, yellow circles represent the ending nodes of important paths, while the green circles represent nodes awaiting sampling. DDLR improves the model's performance by selecting the most relevant paths and edges through path-based and edge-based sampling at each step.

### 4.1 PATH SAMPLING

**Important Paths for Reasoning**. Given a query relation and a lot of entities, only some of the paths between the entities are important for answering the query. Take Fig 1 as an example, it is clear that only evidence path $Father(a, b) \wedge Wife(b, d)$ can lead to the deduction that triplet $Mather(a, d)$ holds true. Formally, we define $\mathcal{P}_{u \to v|q} \subseteq \mathcal{P}_{u \to v}$ to be the set of paths from $u$ to $v$ that is important to the query relation $q$.

$$\boldsymbol{h}_q(u, v) = \bigoplus_{P \in \mathcal{P}_{u \to v}} \boldsymbol{h}_q(P) \approx \bigoplus_{P \in \mathcal{P}_{u \to v|q}} \boldsymbol{h}_q(P) \tag{2}$$

In other words, any path $P \in \mathcal{P}_{u \to v} \setminus \mathcal{P}_{u \to v|q}$ with respect to $\boldsymbol{h}_q(u, v)$ is considered as noise, meaning it is not important to obtain the right answer. Therefore, if we calculate the representation $\boldsymbol{h}_q(u, v)$ using only the important paths $\mathcal{P}_{u \to v|q}$, we can enhance the model's performance.

**Path Scoring**. Given a query of the form $(u, q, ?)$, our aim is to identify the important paths $\mathcal{P}_{u \to v|q}$ for all entities $v$ in the knowledge graph. Nevertheless, extracting these important paths from the extensive set $\mathcal{P}_{u \to v}$ is highly challenging due to its high complexity. So we employ an iterative node selection process to identify the paths that cover the important paths.

$$\hat{\mathcal{P}}^{(0)}_{u \to v|q} = \{(u, \text{selfloop}, v)\} \quad \text{if } u = v \text{ else } \varnothing \tag{3}$$

$$\hat{\mathcal{P}}^{(t)}_{u \to v|q} = \bigcup_{x \in n^{(t-1)}_{uq}(\hat{\mathcal{P}}^{(t-1)}_{u \to x|q}), (x,r,v) \in \mathcal{N}(x)} \{P + \{(x, r, v)\} | P \in \hat{\mathcal{P}}^{(t-1)}_{u \to x|q}\} \tag{4}$$

where $\hat{\mathcal{P}}^{(t)}_{u \to v|q}$ computed by the above iteration is a superset of the important paths $\mathcal{P}^{(t)}_{u \to v|q}$ of length $t$, $n^{(t-1)}_{uq}(\hat{\mathcal{P}}^{(t-1)}_{u \to x|q})$ selects ending nodes of important paths $\mathcal{P}^{(t-1)}_{u \to v|q}$ . $\mathcal{N}(x)$ is the neighborhood of node $x$.

Next, we learn a representation $\boldsymbol{h}^{(t)}_q(u, v)$ based on the paths $\hat{\mathcal{P}}^{(t)}_{u \to v|q}$ of length $t$.

$$\boldsymbol{h}^{(t)}_q(u, v) = \bigoplus_{P \in \hat{\mathcal{P}}^{(t)}_{u \to v|q}} \boldsymbol{h}^{(t)}_q(P) = \bigoplus_{P \in \hat{\mathcal{P}}^{(t)}_{u \to v|q}} \bigotimes_{(x,r,y) \in P} \boldsymbol{w}_q(x, r, y) \tag{5}$$

Then, we utilize a learnable path scoring function $s^{(t)}_{uq}$ to evaluate paths of length $t$. However, evaluating paths from the extensive set $\hat{\mathcal{P}}^{(t)}_{u \to v|q}$ presents a significant challenge due to its exponential size. Therefore, we approximate the evaluation of paths by evaluating the nodes.

$$s^{(t)}_{uq}(\hat{\mathcal{P}}^{(t)}_{u \to v|q}) \approx s^{(t)}_{uq}(\hat{\mathcal{V}}^{(t)}) \tag{6}$$

where $\hat{\mathcal{V}}^{(t)}$ denotes the end nodes of the $\hat{\mathcal{P}}^{(t)}_{u \to v|q}$.

$s^{(t)}_{uq}(x)$ is jointly determined by the current path score $d^{(t)}_q(u, x)$, representing the path score from node $u$ to node $x$, and the remaining path score $r^{(t)}_q(x, v)$, representing the path score from node $x$ to node $v$. Although it is feasible to derive $r^{(t)}_q(x, v)$ from the representation of the remaining path, obtaining the representation is extremely difficult because we do not possess the representation of the answer entity $v$. Hence, we utilize the query relation $\boldsymbol{q}$ as an approximation to get the representation of the remaining path.

$$\boldsymbol{r}^{(t)}_q(x, v) = \boldsymbol{h}^{(t)}_q(u, x) \otimes g([\boldsymbol{h}^{(t)}_q(u, x), \boldsymbol{q}]) \tag{7}$$

where $\boldsymbol{r}^{(t)}_q(x, v)$ denotes the representation of the remaining path, $g(\cdot)$ is a feed-forward network and $[\cdot, \cdot]$ concatenates two representations, $\otimes$ denotes element-wise multiplication.

To achieve a more precise evaluation of paths, it is necessary to take into account the scores of previous nodes within the path.

$$s^{(0)}_{uq}(u) = 0 \tag{8}$$

$$s^{(t)}_{uq}(x) = r^{(t)}_q(x, v) \oplus d^{(t)}_q(u, x) = \quad \sigma(f(\boldsymbol{r}^{(t)}_q(x, v))) \oplus \bigoplus_{(\hat{x}, r, x) \in \mathcal{N}(x)} s^{(t-1)}_{uq}(\hat{x}) \tag{9}$$

where $\sigma$ is the sigmoid function, $f(\cdot)$ is a feed-forward network. $d^{(t)}_q(u, x)$ incorporates the scores of previous nodes within the path.

Finally, we simply pick the top-K paths (i.e., nodes) according to the current scoring function $s^{(t)}_{uq}(x)$.

$$n^{(t)}_{uq}(\hat{\mathcal{P}}^{(t)}_{u \to x|q}) = n^{(t)}_{uq}(\hat{\mathcal{V}}^{(t)}) = \underset{N' \subseteq \hat{\mathcal{V}}^{(t)}, |N'| = K}{\arg\max} \sum_{x \in N'} s^{(t)}_{uq}(x) \tag{10}$$

In order to perform reasoning, we need to compute the representation $\boldsymbol{h}_q^{(t)}(u, v)$ based on the important paths. Then Eq 5 is modified as follows:

$$\boldsymbol{h}_q^{(0)}(u, v) = \boldsymbol{q} \quad \text{if } u = v \text{ else } \boldsymbol{0} \tag{11}$$

$$\boldsymbol{h}_q^{(t)}(u, v) = \bigoplus_{x \in n_{uq}^{(t-1)}(\hat{\mathcal{V}}^{(t-1)}), (x,r,v) \in \mathcal{N}(x)} \boldsymbol{h}_q^{(t-1)}(u, x) \otimes \boldsymbol{w}_q(x, r, v) \tag{12}$$

where $\boldsymbol{q}$ denotes the representation of relation $q$.

## 4.2 EDGE SAMPLING

**Bernoulli Sampling Based On Single Rules**. In addition to investigating the impact of important paths for reasoning, we also explore the influence of relations (i.e., edges) in the reasoning process. For instance, as depicted in Fig. 1, we can observe that the relations within evidence paths exhibit a higher degree of relevance to the target relation compared to those in non-evidence paths. This observation motivates us to sample relations with higher relevance, aiming to assist the model in selecting the most pertinent paths during reasoning. We consider a direct approach to sample important relations by randomly selecting relations from the graph. Formally, we sample a modified subset $\widetilde{\mathcal{R}}^{(t)}$ from the candidate relations $\hat{\mathcal{R}}^{(t)}$ with certain probabilities at the t-hop.

$$\widetilde{\mathcal{R}}^{(t)} = \{r \mid r \in \hat{\mathcal{R}}^{(t)}, Bern(p_{rq}^{(t)}) = 1\} \tag{13}$$

where $p_{rq}^{(t)}$ is the probability of relevance between $r$ and $q$, $Bern$ represents a Bernoulli distribution, $\hat{\mathcal{R}}^{(t)}$ denotes the set of candidate relations, take Fig 1 as an example, $\{Mother, Wife, Friend\}$ is the candidate relation set for the model at the second hop. $\widetilde{\mathcal{R}}^{(t)}$ is then used as the relation set in the generated view. $p_{rq}^{(t)}$ should reflect the importance of the relation $r$, depending on which the model can discard irrelevant relations while retaining those most relevant to the task.

Following logical notations, we denote potential relevance between relation $r_1$ and relation $r_2$ using a single rule $r_1 \Rightarrow r_2$. To quantify the degree of such relevance, we define the confidence of a single rule as follows:

$$\mathcal{C}(r_1 \Rightarrow r_2) = \frac{\sum_{t \in \mathcal{E}} \mathbb{1}(r_1 \in \mathbf{E}_r(t) \wedge r_2 \in \mathbf{E}_r(t))}{\sum_{t \in \mathcal{E}} \mathbb{1}(r_1 \in \mathbf{E}_r(t))} \tag{14}$$

where the function $\mathbb{1}(x)$ equals 1 when $x$ is true and 0 otherwise, $\mathbf{E}_r$ extracts relations from the triplets. As an empirical statistic over the entire KG, $\mathcal{C}(r_1 \Rightarrow r_2)$ is larger if more triplets with relation $r_1$ also have $r_2$.

Next, we calculate the probability of each relation based on its confidence value. The probabilities can then be obtained after a normalization step that transforms the values into possibilities, which is defined as

$$p_{rq}^{(t)} = \min\left(\frac{\mathcal{C}_{max}^{(t)} - \mathcal{C}(r \Rightarrow q)}{\mathcal{C}_{max}^{(t)} - \mathcal{C}_{avg}^{(t)}} \cdot p_e, p_\tau\right) \tag{15}$$

where $p_e$ is a hyperparameter that controls the overall probability of important relations, $\mathcal{C}_{max}^{(t)}$ and $\mathcal{C}_{avg}^{(t)}$ represent the maximum and average values, respectively, within the confidence set $\mathcal{C}^{(t)}$ associated with $\widetilde{\mathcal{R}}^{(t)}$. and $p_\tau$ is a truncation probability used to cut off probability values because extremely low probabilities can result in the loss of important relations.

### 4.3 COMBINATION

Path-based sampling allows the model to select the important paths for reasoning, while edge-based sampling selects important edges. By employing both types of sampling, the model achieves effective denoising for knowledge graph completion. Then, we modify Eq 12 as follows:

$$
\boldsymbol{h}_q^{(t)}(u,v) = \bigoplus_{x \in n_{uq}^{(t-1)}(\hat{\mathcal{V}}^{(t-1)}), r \in \widetilde{\mathcal{R}}^{(t-1)}, (x,r,v) \in \mathcal{N}(x)} \boldsymbol{h}_q^{(t-1)}(u,x) \otimes \boldsymbol{w}_q(x,r,v) \tag{16}
$$

Eq. 16 encourages the model to select important paths for reasoning. A pseudo-code of the algorithm is illustrated in the appendix.

### 4.4 LOSS FUNCTION

We train the model to score positive triples higher than the negative triples using a multi-class log-loss followingLacroix et al. (2018). The loss function $\mathcal{L}$ is

$$
f(u,r,v) = \mathbf{w}_s^T \boldsymbol{h}_q^{(L)}(u,v) \tag{17}
$$

$$
\mathcal{L} = \sum_{(u,r,v) \in \mathcal{E}_{train}} \left( -f(u,r,v) + \log(\sum_{\forall x \in \mathcal{E}} \exp(f(u,r,x))) \right), \tag{18}
$$

where $\mathbf{w}_s \in \mathbb{R}^d$ is a weight parameter, $\boldsymbol{h}_q^{(L)}(u,v)$ represents the embedding of entity $v$ at step L, $\mathcal{E}_{train}$ denotes the set of the positive triples $(u,r,v)$.

## 5 EXPERIMENTS

### 5.1 EXPERIMENT SETUP

**Datasets.** We conducted experiments on three widely recognized datasets: WN18RR(Dettmers et al., 2018), FB15k-237 (Toutanova et al., 2015a), and Nell-995 (Xiong et al., 2017), which are commonly employed for transductive relation prediction tasks. To assess inductive relation prediction, we followed the methodology of Teru et al. (Teru et al., 2020), who created four variations of each dataset. Each inductive dataset consists of a distinct train set and test set, ensuring that their entity compositions do not overlap. For further details, please refer to the supplement.

**Baselines.** In the inductive setting, methods that involve learning entity embeddings during training are not applicable. Consequently, we conduct a comparative analysis with path-based approaches that focus on acquiring sequential rules, including RuleN(Meilicke et al., 2018), NeuralLP(Yang et al., 2017), and DRUM(Sadeghian et al., 2019). Furthermore, we include GNN-based methods such as GraIL(Teru et al., 2020), NBFNet (Zhu et al., 2021), RED-GNN(Zhang & Yao, 2022), Adaprop (Zhang et al., 2023a), and GraPE (Wang et al., 2023) as our baseline models.

**Details.** For the DDLR model, we perform hyperparameter tuning on a variety of settings, including the learning rate within the range of $[10^{-4}, 10^{-2}]$, weight decay within the range of $[10^{-5}, 10^{-2}]$, dropout rate within the range of $[0, 0.3]$, batch size chosen from $\{5, 10, 20, 50, 100\}$, dimension $d$ chosen from $\{32, 48, 64, 96\}$, $d_{att}$ for attention chosen from $\{3, 5\}$, layer $L$ chosen from $\{3, 4, 5, 6, 7\}$, the number of sampled entities $K$ among $\{50, 100, 150, 200, 250, 300\}$, activation function $\delta$ chosen from $\{identity, tanh, ReLU\}$, probability multiplier $p_e$ chosen from $\{0.3, 0.4, 0.5, 0.6, 0.7, 0.8\}$ and cut-off probability $p_\tau$ chosen from $\{0.1, 0.2, 0.3, 0.4, 0.5\}$. We adopt Adam(Kingma & Ba, 2014) as the optimizer. Please refer to the supplement for more detailed information.

**Results.** The performance comparison presented in Table 1 underscores that, despite its inductive reasoning capabilities, GraIL falls notably short of RED-GNN, NBFNet, Adaprop, and GraPE. The patterns acquired by GraIL exhibit limited generalization to unseen knowledge graphs. In contrast,

Table 1: The performance of inductive reasoning on three datasets, where the best results are in **bold** and the second best results are underlined.

| metrics | methods | WN18RR | | | | FB15k237 | | | | NELL-995 | | | |
|---|---|---|---|---|---|---|---|---|---|---|---|---|---|
| | | V1 | V2 | V3 | V4 | V1 | V2 | V3 | V4 | V1 | V2 | V3 | V4 |
| MRR(%) | RuleN | 66.8 | 64.5 | 36.8 | 62.4 | 36.3 | 43.3 | 43.9 | 42.9 | 61.5 | 38.5 | 38.1 | 33.3 |
| | Neural LP | 64.9 | 63.5 | 36.1 | 62.8 | 32.5 | 38.9 | 40.0 | 39.6 | 61.0 | 36.1 | 36.7 | 26.1 |
| | DRUM | 66.6 | 64.6 | 38.0 | 62.7 | 33.3 | 39.5 | 40.2 | 41.0 | 62.8 | 36.5 | 37.5 | 27.3 |
| | GraIL | 62.7 | 62.5 | 32.3 | 55.3 | 27.9 | 27.6 | 25.1 | 22.7 | 48.1 | 29.7 | 32.2 | 26.2 |
| | NBFNet | 68.4 | 65.2 | 42.5 | 60.4 | 30.7 | 36.9 | 33.1 | 30.5 | 58.4 | 41.0 | 42.5 | 28.7 |
| | RED-GNN | 70.1 | 69.0 | 42.7 | 65.1 | 36.9 | 46.9 | 44.5 | 44.2 | 63.7 | 41.9 | 43.6 | 36.3 |
| | AdaProp | 73.3 | 71.5 | 47.4 | 66.2 | 31.0 | 47.1 | 47.1 | 45.4 | 64.4 | 45.2 | 43.5 | 36.6 |
| | GraPE | 74.2 | 70.7 | 47.2 | 65.3 | 41.5 | 48.8 | 48.1 | 47.0 | 77.7 | 49.4 | 45.0 | 38.3 |
| | DDLR | 79.4 | 79.8 | 52.0 | 76.4 | 48.8 | 49.7 | 48.8 | 46.3 | 81.1 | 48.2 | 46.5 | 40.6 |
| Hit@1 (%) | RuleN | 63.5 | 61.1 | 34.7 | 59.2 | 30.9 | 34.7 | 34.5 | 33.8 | 54.5 | 30.4 | 30.3 | 24.8 |
| | Neural LP | 59.2 | 57.5 | 30.4 | 58.3 | 24.3 | 28.6 | 30.9 | 28.9 | 50.0 | 24.9 | 26.7 | 13.7 |
| | DRUM | 61.3 | 59.5 | 33.0 | 58.6 | 24.7 | 28.4 | 30.8 | 30.9 | 50.0 | 27.1 | 26.2 | 16.3 |
| | GraIL | 55.4 | 54.2 | 27.8 | 44.3 | 20.5 | 20.2 | 16.5 | 14.3 | 42.5 | 19.9 | 22.4 | 15.3 |
| | NBFNet | 59.2 | 57.5 | 30.4 | 57.4 | 19.0 | 22.9 | 20.6 | 18.5 | 50.0 | 27.1 | 26.2 | 23.3 |
| | RED-GNN | 65.3 | 63.3 | 36.8 | 60.6 | 30.2 | 38.1 | 35.1 | 34.0 | 52.5 | 31.9 | 34.5 | 25.9 |
| | AdaProp | 66.8 | 64.2 | 39.6 | 61.1 | 19.1 | 37.2 | 37.7 | 35.3 | 52.2 | 34.4 | 33.7 | 24.7 |
| | DDLR | 70.9 | 70.2 | 42.7 | 67.2 | 38.2 | 37.9 | 39.4 | 36.1 | 73.6 | 35.1 | 35.0 | 28.9 |
| Hit@10 (%) | RuleN | 73.0 | 69.4 | 40.7 | 68.1 | 44.6 | 59.9 | 60.0 | 60.5 | 76.0 | 51.4 | 53.1 | 48.4 |
| | Neural LP | 77.2 | 74.9 | 47.6 | 70.6 | 46.8 | 58.6 | 57.1 | 59.3 | 87.1 | 56.4 | 57.6 | 53.9 |
| | DRUM | 77.7 | 74.7 | 47.7 | 70.2 | 47.4 | 59.5 | 57.1 | 59.3 | 87.3 | 54.0 | 57.7 | 53.1 |
| | GraIL | 76.0 | 77.6 | 40.9 | 68.7 | 42.9 | 42.4 | 42.4 | 38.9 | 56.5 | 49.6 | 51.8 | 50.6 |
| | NBFNet | 59.2 | 57.5 | 30.4 | 57.4 | 19.0 | 22.9 | 20.6 | 18.5 | 50.0 | 27.1 | 26.2 | 23.3 |
| | RED-GNN | 79.9 | 78.0 | 52.4 | 72.1 | 48.3 | 62.9 | 60.3 | 62.1 | 86.6 | 60.1 | 59.4 | 55.6 |
| | AdaProp | 86.6 | 83.6 | 62.6 | 75.5 | 55.1 | 65.9 | 63.7 | 63.8 | 88.6 | 65.2 | 61.8 | 60.7 |
| | DDLR | 98.0 | 98.8 | 69.3 | 95.6 | 64.5 | 71.0 | 63.8 | 64.5 | 91.5 | 69.0 | 67.6 | 60.3 |

DDLR consistently outperforms its counterparts across various datasets and partition variations, frequently securing the best or second-best rankings. This highlights DDLR's capacity to enhance inductive reasoning through a dual denoising approach, effectively filtering out irrelevant paths while retaining those most relevant to the task. For more experimental analysis, please refer to the supplement.

## 5.2 ABLATION STUDIES

**Effectiveness of each component.** To analyze the contributions of each component within DDLR to its overall performance, we conducted a series of ablation experiments aimed at validating the effectiveness of these model components. Firstly, we investigated the impact of removing the edge-based sampling module, referred to as DDLR-w.o.-edge. Specifically, we retained only the path-based sampling module. Since the model does not consider the correlations between relations in the knowledge graph (i.e., single rules), inductive reasoning can be affected by the noise introduced by irrelevant edges, potentially resulting in a performance decrease. Secondly, we retained only the path-based sampling module to observe the influence of path-based sampling, denoted as DDLR-w.o.-path. Due to the model's limited ability to accurately evaluate reasoning paths, it may be influenced by irrelevant reasoning paths, thereby reducing its generalization capacity. Lastly, to validate the higher credibility of triplet-level single rules, we used entity-level single rules as a reference, referred to as DDLR-w.o.-triplet. The experimental results are depicted in Table 2. These ablation experiments provide valuable insights into the pivotal roles played by individual components of DDLR in enhancing its inductive reasoning capabilities.

**Path-Scoring Function**. The path scoring is composed of both the current path score and the remaining path score. To examine the influence of each component, we conducted a simple ablation experiment: "DDLR-w.o.-current" denotes the removal of the current path score, while "DDLR-w.o.-remain" denotes the removal of the remaining path score. The experimental results, as shown in Table 3, indicate that the absence of any of these subcomponents leads to a decrease in model performance. This indirectly confirms the importance of both the current path score and the remaining path score.

Table 2: Ablation studies of each component. 'DDLR-w.o.-edge' denotes the model without edge sampling. 'DDLR-w.o.-path' denotes the model without path sampling. 'DDLR-w.o.-triplet' denotes entity-level single rules.

| methods | WN18RR(V4) | FB15k237(V4) | NELL-995(V4) |
|---|---|---|---|
| | MRR(%) | MRR(%) | MRR(%) |
| DDLR-w.o.-edge | 73.3 | 45.2 | 35.6 |
| DDLR-w.o.-path | 76.2 | 44.6 | 38.8 |
| DDLR-w.o.-triplet | 74.1 | 43.4 | 30.8 |
| DDLR | 76.4 | 46.3 | 40.6 |

Table 3: Ablation study results of path-scoring function components. 'DDLR-w.o.-current' refers to the path-scoring functions excluding the current path score. 'DDLR-w.o.-remain' refers to the path-scoring functions excluding the remaining path score.

| methods | WN18RR(V2) | FB15k237(V3) | NELL-995(V1) |
|---|---|---|---|
| | MRR(%) | MRR(%) | MRR(%) |
| DDLR-w.o.-current | 75.9 | 48.4 | 77.0 |
| DDLR-w.o.-remain | 75.9 | 48.2 | 76.0 |
| DDLR | 79.4 | 48.8 | 81.1 |

**Relevance Between Relations.** To verify the reliability of single rules, we compared them with other methods measuring relation relevance, as shown in Table 4. The results demonstrate a clear advantage of single rules. It also indicates that single rules at the triplet level possess higher credibility, which aids the DDLR model in filtering out irrelevant edges.

Table 4: Comoarisons of different methods to measure the relation relevance. The term "Cosine" represents the computation of cosine similarity between relations, while "KL_DIV" and "JS_DIV" respectively denote the computation of KL divergence and JS divergence between relations.

| datasets | Ours | Cosine | KL_DIV | JS_DIV |
|---|---|---|---|---|
| | MRR(%) | MRR(%) | MRR(%) | MRR(%) |
| NELL-995(V1) | 81.1 | 70.6 | 64.2 | 70.3 |
| WN18RR(V1) | 79.4 | 79.3 | 78.1 | 78.2 |

## 6 CONCLUSION

The DDLR framework proposed in our work effectively enhances knowledge graph completion by integrating dual denoising techniques operating at both the path and edge levels which allows for the efficient filtering and selection of relevant information, resulting in improved accuracy and efficiency in knowledge graph completion tasks. The path-scoring mechanism prioritizes important paths during the reasoning process, reducing the influence of noise from less relevant paths. Furthermore, the edge-based denoising technique further refines the selection of paths. These two components work in concert to enhance the overall performance of the model. Despite the excellent performance, DDLR has some limitations. Its current evaluation scope is primarily centered on reasoning for missing triplets, a relatively straightforward task. Yet, addressing the complexities associated with multi-hop queries presents a more challenging endeavor. In the forthcoming research, we plan to investigate alternative learning strategies and extend the application of DDLR to these intricate reasoning

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
