# A APPENDIX

## A.1 DATASETS & EVALUATION

In Table 5, we present an overview of the dataset statistics related to inductive knowledge graph reasoning. In the inductive scenario, we adhere to the approach outlined in Zhang & Yao (2022), where each target tail (or head) entity is ranked in comparison to the remaining negative entities. Our reported metrics include the mean reciprocal rank (MRR) of the rankings and the hit rate (H@K) for the top K rankings.

Table 5: Statistics of inductive benchmarks. We use REL and ENT and TR to denote the number of relations, entities, and triplets, respectively

| DataSets | | WN18RR | | | FB15k-237 | | | NELL-995 | | |
| | | REL | ENT | TR | REL | ENT | TR | REL | ENT | TR |
|---|---|---|---|---|---|---|---|---|---|---|
| v1 | train | 9 | 2746 | 6678 | 183 | 2000 | 5226 | 14 | 10915 | 5540 |
| | test | 9 | 922 | 1991 | 146 | 1500 | 2404 | 14 | 225 | 1034 |
| v2 | train | 10 | 6954 | 18968 | 203 | 3000 | 12085 | 88 | 2564 | 10109 |
| | test | 10 | 2923 | 4863 | 176 | 2000 | 5092 | 79 | 4937 | 5521 |
| v3 | train | 11 | 12078 | 32150 | 218 | 4000 | 22394 | 142 | 4647 | 20117 |
| | test | 11 | 5084 | 7470 | 187 | 3000 | 9137 | 122 | 4921 | 9668 |
| v4 | train | 9 | 3861 | 9842 | 222 | 5000 | 33916 | 77 | 2092 | 9289 |
| | test | 9 | 7208 | 15157 | 204 | 3500 | 14554 | 61 | 3294 | 8520 |

## A.2 HYPER-PARAMETERS

We provide the hyper-parameters of $L$, $K$, $p_e$, $p_\tau$ and batch size in Table 6.

Table 6: Hyperparameter configurations of DDLR on different datasets.

| Hyper-parameters | WN18RR | | | | FB15k237 | | | | NELL-995 | | | |
| | V1 | V2 | V3 | V4 | V1 | V2 | V3 | V4 | V1 | V2 | V3 | V4 |
|---|---|---|---|---|---|---|---|---|---|---|---|---|
| $L$ | 3 | 3 | 7 | 3 | 7 | 3 | 7 | 5 | 3 | 3 | 3 | 7 |
| $K$ | 150 | 50 | 100 | 300 | 300 | 250 | 300 | 300 | 50 | 300 | 300 | 100 |
| $p_e$ | 0.5 | 0.3 | 0.3 | 0.6 | 0.3 | 0.7 | 0.3 | 0.4 | 0.5 | 0.4 | 0.4 | 0.8 |
| $p_\tau$ | 0.5 | 0.5 | 0.5 | 0.5 | 0.5 | 0.5 | 0.5 | 0.5 | 0.3 | 0.4 | 0.4 | 0.5 |
| batch size | 100 | 50 | 100 | 10 | 20 | 10 | 20 | 20 | 10 | 100 | 10 | 20 |

## A.3 EFFECTIVENESS OF CUT-OFF PROBABILITY

Fig 3 illustrates the impact of different values of $p_\tau$. The results indicate that the choice of the cut-off probability may have a noticeable impact on performance. Around a specific cut-off probability value, the model may exhibit optimal performance, while in the vicinity of other cut-off probability, performance may decline. Therefore, in practical applications, careful consideration should be given to selecting the cut-off probability value to optimize the model's performance on specific tasks and datasets.

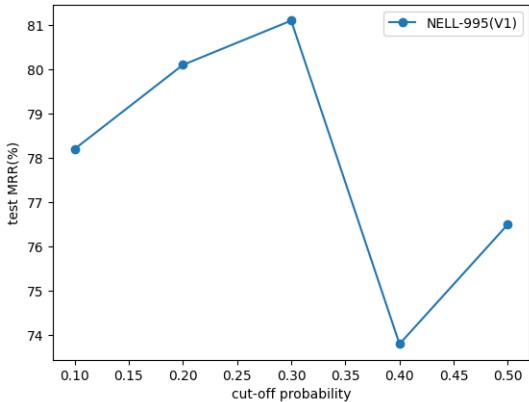

Figure 3: Ablation study with different $p_\tau$

## A.4 FULL ALGORITHM

The full procedure is shown in Algorithm 1. In line 3, we initialize embeddings for all nodes and obtain the ending nodes of the important paths with a length of $t - 1$ through the path scoring function and the top-K strategy in line 6. Next, DDLR conducts edge sampling in line 8. Finally, we obtain the representation $\boldsymbol{h}_q^{(t)}(u, v)$ in line 12 and path scores $s_{uq}^{(t)}(x)$ in line 13.

---

**Algorithm 1:**

---

    **Input:** head entity $u$, query relation $q$, iterations $L$

    **Output:** $\boldsymbol{h}_q^{(L)}(u, v)$ for $v \in \mathcal{V}$

1  $\hat{\mathcal{V}}^{(0)} = \{u\}$

2  **for** $v \in \mathcal{V}$ **do**

3    $\boldsymbol{h}_q^{(0)}(u, v) = \boldsymbol{q}$     if u=v else $\boldsymbol{0}$

4  **end**

5  **for** $t = 1$ to $L$ **do**

6    $\chi^{(t-1)} = \text{TopK}(s_{uq}^{(t)}(x) \mid x \in \hat{\mathcal{V}}^{(t-1)})$

7    $\mathcal{E}^{(t)} = \bigcup_{x \in \chi^{(t-1)}} (x, r, v) \in \mathcal{N}(x)$

8    Compute $\widetilde{\mathcal{R}}^{(t)}$ with Eq 13

9    $\mathcal{E}^{(t)} = \{(x, r, v) \mid (x, r, v) \in \mathcal{E}^{(t)}, r \in \widetilde{\mathcal{R}}^{(t)}\}$

10  $\hat{\mathcal{V}}^{(t)} = \bigcup_{(x,r,v) \in \mathcal{E}^{(t)}} \{v\}$

11    **for** $v \in \hat{\mathcal{V}}^{(t)}$ **do**

12      Compute $\boldsymbol{h}_q^{(t)}(u, v)$ with Eq 16

13      Compute $s_{uq}^{(t)}(x)$ with Eq 9

14    **end**

15  **end**

16  **return** $\boldsymbol{h}_q^{(L)}(u, v)$

---