# OpenReview forum: "DUAL DENOISING LOGICAL REASONING FOR INDUCTIVE KNOWLEDGE GRAPH COMPLETION"
_ICLR.cc/2024/Conference — Submitted to ICLR 2024_

### Official Review · Reviewer_QaFK · 2023-10-26

**Soundness:** 1 poor
**Presentation:** 1 poor
**Contribution:** 1 poor
**Rating:** 1
**Confidence:** 5

**Summary:**

This paper focuses on path-based GNNs for knowledge graph reasoning and proposes two modifications to existing models: (1) sampling of important paths (edges) in the graph to reduce the number of visited edges; (2) further sub-sampling of edges based on pre-computed counting features of overlapping relations.

**Strengths:**

N/A

**Weaknesses:**

This path sampling section of DDLR is a blatant plagiarism of the work “A\*Net: A Scalable Path-based Reasoning Approach for Knowledge Graphs” [1] available on arxiv since June 2022, I’ll therefore refer to the original 2022 work as “the A\*Net paper” or “A\*Net”, and to this ICLR submission as “DDLR”.

This submission (DDLR) copies almost verbatim several sections of the A\*Net paper including images and equations without any reference or citation of the original work:
* Section 1: Figure 1 in DDLR is a re-colored version of Figure 2(a) in A*Net.
* Section 3 (Preliminary): The Knowledge Graph Completion paragraph is a copy of the same paragraph in the A*Net with some words replaced by synonyms, eg,
    * [A*Net] “Each fact is a triplet $(x,r,y) \in \mathcal{V} \times \mathcal{R} \times \mathcal{V}$, which indicates a relation $r$ from entity $x$ to entity $y$.”
    * [DDLR] “Each fact is a triplet $(x,r,y) \in \mathcal{V} \times \mathcal{R} \times \mathcal{V}$, signifying the presence of relation $r$ from entity $x$ to entity $y$.”
* Section 3 (Preliminary): The Knowledge Graph Completion paragraph
    * [A*Net] The task of knowledge graph reasoning aims at answering queries like $(u,q,?)$ or $(?,q,u)$. … we assume the query is $(u,q,?)$, since $(?,q,u)$ equals to $(u,q^{-1},?)$ with $q^{-1}$ being the inverse of $q$.
    * [DDLR] Knowledge graph completion is a task focused on predicting answers for queries such as $(u,q,?)$ or equivalently $(u,q^{-1},?)$, where $q^{-1}$ represents the inverse relation of $q$. **Here the authors make a mistake because $(u,q^{-1},?)$ is not equivalent to $(u,q,?)$ but instead is equivalent to the head prediction task to $(?,q,u)$.**
* Section 3 (Preliminary) -> Path-based Approaches: almost verbatim copy of the text, equation 1, and the same notation from the A\*Net paragraph Path-based Methods.
* Section 4.1 (Path Sampling) -> Important Paths for Reasoning: the whole paragraph is a verbatim copy of Section 3.1 (Important Paths for Reasoning) from the A*Net paper with non-sensical word replacements, eg
    * [A*Net] Given a query relation and a pair of entities, only some of the paths between the entities are important for answering the query.
    * [DDLR] Given a query and **a lot of** entities, only some of the paths between the entities are important for answering the query.
* Section 4.1 (Path Sampling) -> Path Scoring includes copy-paste excerpts from A\*Net’s paragraphs Iterative Computation of Important Paths, Reasoning with A* Algorithm, and Neural Priority function. Particularly,
    * Equations 3,4,5,7 (DDLR) are the same as Equations 6,8,9,10 from A*Net with the same notation and descriptions
    * The priority function $s_{uq}(x)$ is named as “learnable path scoring function” in DDLR instead of “neural priority function” in A\*Net

Even if we omit the borrowed A\*Net content from the list of contributions, the paper boils down to pre-computing co-occurences of all pairs of relations and tuning the threshold manually for each dataset - this is not a substantial algorithmic contribution per se.

Overall, I believe the paper should be rejected as it clearly violates the ICLR Code of Ethics (that authors are supposed to accept upon the submission) and escalated to the AC / SAC / PC level for further investigation.


[1] https://arxiv.org/abs/2206.04798

**Questions:**

N/A

**Details Of Ethics Concerns:**

Suspected plagiarism of the A*Net paper that has been on arxiv since June 2022.

---

### Official Review · Reviewer_m7i6 · 2023-10-30

**Soundness:** 1 poor
**Presentation:** 2 fair
**Contribution:** 1 poor
**Rating:** 1
**Confidence:** 4

**Summary:**

This paper addresses the issue of noise in message-passing Graph Neural Networks (GNNs) used in inductive knowledge graph completion, where irrelevant entities and edges introduce noise that accumulates exponentially, hindering model performance. A dual denoising logical reasoning (DDLR) framework is proposed that integrates path-based and edge-based sampling mechanisms. This framework employs a path-scoring mechanism to evaluate and remove irrelevant paths and leverages rules within the knowledge graph to exclude irrelevant edges, aiming for comprehensive denoising. Experiments conducted on three benchmark datasets show that the DDLR framework achieves state-of-the-art performance, demonstrating its effectiveness in enhancing logical reasoning in GNN-based knowledge graph reasoning.

**Strengths:**

1.	The proposed method can effectively enhance knowledge graph completion by integrating dual denoising techniques operating at both the path and edge levels.
2.	The authors conducted extensive experiments to demonstrate the effectiveness of the method, which presented the performance for each component from various perspectives.

**Weaknesses:**

1.	The paper is not organized clearly, which is not friendly for understanding. Specifically, there are lack the detailed discussion about related work [1].
2. This paper should be sent to the ICLR ethical board review committee on the subject of scientific misconduct and there are some ethics problems in this paper according to the A*Net paper [1].
[1] A*Net: A Scalable Path-based Reasoning Approach for Knowledge Graphs

**Questions:**

Please refer to the weaknesses.

**Details Of Ethics Concerns:**

There are some ethics problems in this paper according to the A*Net paper [1].
[1] A*Net: A Scalable Path-based Reasoning Approach for Knowledge Graphs

---

### Official Review · Reviewer_AW5P · 2023-11-01

**Soundness:** 2 fair
**Presentation:** 2 fair
**Contribution:** 1 poor
**Rating:** 5
**Confidence:** 4

**Summary:**

This paper proposes a message-passing mechanism for inductive knowledge graph completion.

Given the fact that this method is highly similar to the existing A*Net, I don't think the contribution was properly justified in the current version.

**Strengths:**

The performance is reasonably good, given the success of A*Net and GraPE. Additionally to its predecessors, section 4.2 is somewhat true.

**Weaknesses:**

The only part of the model that is a significant difference between this work and A*Net is Section 4.2.

I don't see any excuses for not comparing this work with A*Net, especially since this paper includes a more recent arxived baseline GraPE.

**Questions:**

What is the difference of this paper against A*Net? How do the potential differences lead to the performance gain/loss?

---

### Official Review · Reviewer_PQGA · 2023-11-04

**Soundness:** 3 good
**Presentation:** 3 good
**Contribution:** 2 fair
**Rating:** 3
**Confidence:** 4

**Summary:**

The paper proposes a new framework called Dual Denoising Logical Reasoning (DDLR) for inductive knowledge graph completion. Current GNN-based methods suffer from noise accumulation when propagating messages freely through graphs. This impacts performance. DDLR employs dual denoising through path-based and edge-based sampling to remove irrelevant information. For path sampling, a novel path scoring mechanism is proposed to evaluate path importance for selecting relevant paths. For edge sampling, single rules extracted from the graph are used to sample relevant edges and filter out irrelevant ones. Experiments on 3 datasets show DDLR outperforms state-of-the-art methods for inductive knowledge graph completion.

**Strengths:**

* The dual denoising approach combining path and edge sampling is innovative. It allows comprehensive filtering of noise from both entities and relations.

* Extensive experiments validate the effectiveness of DDLR, demonstrating superior accuracy over state-of-the-art baselines.

**Weaknesses:**

A few sections of the paper are nearly identical or highly similar to the A\*Net paper [1]. However, the A\*Net paper is not cited anywhere. This raises ethical concerns, and puts the novel contributions of this work into question. The authors need to clearly distinguish their contributions compared to A\*Net and other prior works through proper citation and explanation. As presented, the incremental novelty is unclear since the core ideas appear derived from A\*Net without attribution. This is a major weakness.

[1] A*Net: A Scalable Path-based Reasoning Approach for Knowledge Graphs

**Questions:**

See the section of weakness above.

---

### Meta-Review · Area_Chair_ksSG · 2023-12-07

**Metareview:**

The paper proposes a new framework called Dual Denoising Logical Reasoning (DDLR) for inductive knowledge graph completion. It integrates path-based and edge-based sampling to achieve comprehensive denoising.  Reviewers noted that the paper has a few sections that are nearly identical or highly similar to the A*Net paper which was not cited. The paper has to address such concerns and clearly distinguish from existing work in the next version.

**Justification For Why Not Higher Score:**

The reviewers raised serious concerns about this paper, especially with regard to the A*Net paper, which the authors did not address.

**Justification For Why Not Lower Score:**

N/A

---

### Decision · Program_Chairs · 2024-01-16

Reject